Tissue-specific evaluation of suitable reference genes for RT-qPCR in the pond snail, Lymnaea stagnalis

http://orcid.org/0000-0003-3395-2377 Young Alexander P. 1 ayoung@stfx.ca
Landry Carmen F. 1
http://orcid.org/0000-0001-9045-381X Jackson Daniel J. 2
http://orcid.org/0000-0003-0971-7588 Wyeth Russell C. 1
1 Department of Biology, St. Francis Xavier University , Antigonish, NS , Canada
2 Department of Geobiology, Georg-August Universität Göttingen , Göttingen , Germany
Brembs Björn
Electronic publication date: 2019 Oct 15
Publication date: 2019
Volume: 7
Electronic Location ID: e7888
Received 2019 Apr 29; Accepted 2019 Sep 13
Copyright: © 2019 Young et al.
Copyright year: 2019
Copyright holder: Young et al.
License: This is an open access article distributed under the terms of the Creative Commons Attribution License, which permits unrestricted use, distribution, reproduction and adaptation in any medium and for any purpose provided that it is properly attributed. For attribution, the original author(s), title, publication source (PeerJ) and either DOI or URL of the article must be cited.
License URL: https://creativecommons.org/licenses/by/4.0/

Keywords: Gene expression, Normalization, Mollusc, Gastropod neurobiology

Funding: Canadian Foundation for Innovation 25196, 17075, 19286 Natural Sciences and Engineering Research Council of Canada RGPIN-2015-04957 Deutsche Forschungsgemeinschaft JA 2108/6-1 Support was received from the Canadian Foundation for Innovation (Grant 25196 to Dr. Cory Bishop, Grant to 17075 Dr. Angela Beye, and Grant 19286 to Dr. Russell Wyeth), the Natural Sciences and Engineering Research Council of Canada (Discovery grant RGPIN-2015-04957 to Dr. Russell Wyeth, CGS-M and a Michael Smith Foreign Study Supplement to Alexander Young), the Deutsche Forschungsgemeinschaft (Grant JA 2108/6-1 to Dr. Daniel Jackson), and St. Francis Xavier University. The funders had no role in study design, data collection and analysis, decision to publish, or preparation of the manuscript.

==============================
Reverse transcription quantitative PCR (RT-qPCR) is a robust technique for the quantification and comparison of gene expression. To obtain reliable results with this method, one or more reference genes must be employed to normalize expression measurements among treatments or tissue samples. Candidate reference genes must be validated to ensure that they are stable prior to use in qPCR experiments. The pond snail (Lymnaea stagnalis) is a common research organism, particularly in the areas of learning and memory, and is an emerging model for the study of biological asymmetry, biomineralization, and evolution and development. However, no systematic assessment of qPCR reference genes has been performed in this animal. Therefore, the aim of our research was to identify stable reference genes to normalize gene expression data from several commonly studied tissues in L. stagnalis as well as across the entire body. We evaluated a panel of seven reference genes across six different tissues in L. stagnalis with RT-qPCR. The genes included: elongation factor 1-alpha, glyceraldehyde-3-phosphate dehydrogenase, beta-actin, beta-tubulin, ubiquitin, prenylated rab acceptor protein 1, and a voltage gated potassium channel. These genes exhibited a wide range of expression levels among tissues. The tissue-specific stability of each of the genes was consistent when measured by the standard stability assessment algorithms: geNorm, NormFinder, BestKeeper, and RefFinder. Our data indicate that the most stable reference genes vary among the tissues that we examined (central nervous system, tentacles, lips, penis, foot, mantle). Our results were generally congruent with those obtained from similar studies in other molluscs. Given that a minimum of two reference genes are recommended for data normalization, we provide suggestions for strong pairs of reference genes for single- and multi-tissue analyses of RT-qPCR data in L. stagnalis.

Background

Lymnaea stagnalis is a freshwater snail with an extensive history of proximate mechanism research. Previous studies have primarily focused on its neurobiology, as well as its endocrinology, immunology, and shell morphology (Chase, 2002; Benjamin, 2008). The central nervous system (CNS) is relatively simple, with approximately 20,000 neurons, many of which are large and easily identifiable, making them amenable for studies of learning, memory, motor pattern generation, neuronal regeneration, and synapse formation (Lukowiak, 2016; Elliott & Susswein, 2002; Chase, 2002; Lukowiak et al., 2003; Getz et al., 2018). Parallel investigations into the neuroendocrine (Koene, 2010; Pirger et al., 2010) and immune (Van Der Knaap, Adema & Sminia, 1993; Gust et al., 2013a) systems of L. stagnalis and other gastropods have improved our understanding of reproductive behavior, immunity and toxicology of L. stagnalis. Additionally, L. stagnalis has been used as a research organism to study the molecular mechanisms that guide shell formation (Boer & Witteveen, 1980; Ebanks, O’Donnell & Grosell, 2010; Hohagen & Jackson, 2013; Jackson, Herlitze & Hohagen, 2016; Herlitze et al., 2018). More recently, L. stagnalis has also been used to study the molecular basis of body asymmetry (Davison et al., 2016; Abe & Kuroda, 2019).

A broad range of traditional techniques have been used in past studies, but modern molecular genetics have yet to be thoroughly incorporated into the study of L. stagnalis. Methods to study the genetics of L. stagnalis such as reverse transcription quantitative PCR (RT-qPCR), in situ hybridization, and transcriptomics (among others) have been established but have not yet been used extensively (Feng et al., 2009; Herlitze et al., 2018). The few studies using RT-qPCR have spanned several topics including chirality, learning, and memory (Sadamoto et al., 2010; Foster, Lukowiak & Henry, 2015; Davison et al., 2016; Korneev et al., 2018; Dong et al., 2018). Additionally, a small number of studies have used in situ hybridization in this species, primarily in sections and whole mounts of the CNS, as well as larvae (Dirks et al., 1989; Boer et al., 1992; Croll & Van Minnen, 1992; Sadamoto et al., 2004). Our goal here is to continue to build the foundation for further molecular genetics studies in L. stagnalis via validation of candidate reference genes for RT-qPCR.

RT-qPCR is a robust technique for the quantification of the mRNA for a specific transcript. It can be used alongside in situ hybridization or other visualization techniques to establish patterns of gene expression in animals. For such experiments, relative quantification against one or more reference genes is the most common choice to compare gene expression across samples (Schmittgen & Livak, 2008; Ruijter et al., 2009). Historically, highly expressed cellular maintenance genes have been haphazardly selected as reference genes for qPCR experiments in many species because expression of such genes was thought to be inherently stable (Kozera & Rapacz, 2013). However, many of these genes have been shown to have unstable expression in several species of plants and animals, highlighting the importance of species-specific validation prior to use (Dheda et al., 2004; Barber et al., 2005; De Jonge et al., 2007; Tong et al., 2009; Eissa et al., 2016). Therefore, validation that candidate reference genes have stable expression is critical for the acquisition of accurate qPCR data and the experimental approaches that incorporate this technique.

To our knowledge, the only previous example of reference gene validation in L. stagnalis is for EF1α in the CNS (Foster, Lukowiak & Henry, 2015). Other experiments have used reference genes without validation, including elongation factor 1-alpha (EF1α) (Gust et al., 2013b; Shimizu et al., 2013), glyceraldehyde-3-phosphate dehydrogenase (GAPDH) (Aleksic & Feng, 2012), beta-actin (ACTB) (Senatore & Spafford, 2012; Hatakeyama et al., 2013; Carter et al., 2015) and beta-tubulin (TUBB) (Bavan et al., 2012; Korneev et al., 2013; Gust et al., 2013b; Flynn et al., 2014; Carter et al., 2015; Benatti et al., 2017). To establish a more rigorous foundation for future research using RT-qPCR, we present an analysis of seven candidate reference genes across six tissues of interest (tentacles, lips, foot, penis, mantle, and CNS) in L. stagnalis. We also provide a combined analysis with data from all tissues to demonstrate which genes are likely to be most stably expressed in whole-body preparations of L. stagnalis.

We investigated the stability of Lst-EF1α, Lst-GAPDH, Lst-ACTB, Lst-TUBB, and Lst-ubiquitin (UBI) as they are commonly employed reference genes. Messenger RNA transcripts encoding the prenylated rab acceptor protein 1 (Rapac1) and a voltage gated potassium channel (VGKC) were also assessed as analogs of these genes have recently been found to be stable in the terrestrial gastropod Cepaea nemoralis (Affenzeller, Cerveau & Jackson, 2018).

Methods

Care of snails

Animal use was consistent with the Canadian Council for Animal Care guidelines. A colony of L. stagnalis was bred and maintained in the animal care facility at StFX. The animals were exposed to a photoperiod matched to natural daylight patterns in Antigonish, Nova Scotia. The water in the animals’ tanks was changed three times per week. The animals were fed fish food and romaine lettuce ad libitum plus sinking protein pellets once per week.

Total RNA extraction

Total RNA was collected from each L. stagnalis tissue of interest. To prepare for RNA isolation, animals were anesthetized in 0.125% 1-phenoxy-2-propanol dissolved in Lymnaea saline for up to 30 min (Wyeth et al., 2009). The animals were dissected to isolate the six tissues of interest: CNS, tentacles, lips, penis, foot, and mantle. The tissues of each type from two animals were combined for each replicate to ensure adequate yields of RNA from each tissue, three replicates were produced with this method. Tissues were flash frozen in liquid nitrogen, shredded with razor blades and added to 500 µL of TRIzol reagent (Thermo Fisher, Waltham, MA, USA). Tissue solutions were thoroughly homogenized with a PowerGen 125 tissue homogenizer (Thermo Fisher, Waltham, MA, USA). Total RNA was extracted from the homogenized tissue via addition of 0.5 volumes of a 24:1 solution of chloroform and isoamyl alcohol. The aqueous layer was separated, and RNA was precipitated with isopropyl alcohol. RNA precipitate was transferred to the spin column of the E.Z.N.A.® Mollusc RNA Kit (Omega Bio-Tek, Norcross, GA, USA) where the RNA was washed and subjected to the on-column DNAse I treatment as per the manufacturer’s instructions. To confirm the effectiveness of the DNase I treatment, PCR was performed with primer sets for Lst-tyrosine hydroxylase (forward: 5′-CCCACGTGTATCGTCATCTTA-3′; reverse: 5′-ATCTTCTCCTCGCAAAACC-3′) and Lst-EF1α with 20 ng of total RNA as a template as this reaction would not work in the absence of genomic DNA (gDNA) contamination. Lst-tyrosine hydroxylase was chosen as one of the gDNA contamination controls as it is frequently amplified by us in all of the tissues of interest as a part of a larger research project on neural development. No amplification was observed for either gene in any of the RNA samples after 40 cycles, thus the RNA samples were judged to be free of gDNA contamination.

Total RNA was quantified in a QuBit 3.0 (Thermo Fisher, Waltham, MA, USA). Yields of 50–200 ng/µL dissolved in 30 µL of molecular grade water were common for roughly 40 mg of tissue mass. A sample of total RNA was measured with a spectrophotometer to confirm a 260:280 ratio of 1.8–2.0. Lastly, five µL of each RNA sample was denatured in five µL of 2× RNA Loading Dye (New England Biolabs, Ipswich, MA, USA) at 75 °C for 10 min and then immediately placed on ice. The RNA samples were loaded on a 1% agarose non-denaturing gel and run alongside a single stranded RNA ladder (New England Biolabs, Ipswich, MA, USA) to verify that the 28S and 18S rRNA bands were intact and the RNA samples were not degraded.

Reverse transcription PCR

Reverse transcription-PCR was performed with iScript Reverse Transcription Supermix for RT-qPCR (Bio-Rad Laboratories, Hercules, CA, USA) which contains a mixture of oligo(dT) primers and random hexamers. For RT-PCR reactions, 500 ng of RNA was added to four µL of iScript Supermix and topped up to 20 µL with molecular grade water as recommended by the manufacturer. RT-PCR took place in a Bio-Rad CFX Connect thermocycler (Bio-Rad Laboratories, Hercules, CA, USA). The RT-PCR program ran at 25 °C for 5 min, 46 °C for 20 min and 95 °C for 1 min.

Design, validation, and optimization of primers for quantitative PCR

Candidate primers intended for qPCR were designed with CLC Main Workbench software (Qiagen, Hilden, Germany). Primers were designed to have a length of 19–23 nucleotides, a melting temperature between 55 and 62 °C, a GC content between 40% and 60% and produce a product no more than 200 bp long. Additionally, the GC content of each primer was kept within 10% of its counterpart, and the melting temperature of each was kept within 3 °C of its counterpart.

Two to six sets of primers were designed for each gene so that optimal primers could be selected for the qPCR reactions. The primers were tested on combined cDNA samples to minimize tissue-specific bias. Melt curves were performed to verify that one product was amplified. If the primers produced a single product, then five µL of the PCR reactions were run on a 2% (w/v) agarose gel at 60 V for roughly 30 min alongside a 50 bp DNA Ladder (New England BioLabs, Ipswich, MA, USA). The agarose gels were analyzed with a Bio-Rad ChemiDoc (Bio-Rad Laboratories, Hercules, CA, USA) to visualize the size of the product. Ultimately, one set of primers was selected for each candidate reference gene based on the quality of the PCR product. The primers for each candidate reference gene are listed in Table 1.

Table 1 Description of all primers used to amplify candidate reference genes in Lymnaea stagnalis.

Primers labeled “For” are forward primers and primers labeled “Rev” are reverse primers, all sequences are written in the 5′ to 3′ direction. The amplification efficiency was determined from each reference gene primer set following RT-qPCR with five 1:5 serial dilutions of total RNA from 100 ng.

Gene	Primer Sequence (5′ to 3′)	Function	Product Size (bp)	Tm (°C)	Efficiency	r2	Accession	
Lst-ACTB	For [AGGCCAACAGAGAAAAGA]
Rev [AGATGCGTACAGAGAGAG]	Cell structure and motility	97	56	2.12	0.999	KX387883	
Lst EF1α	For [ACCACAACTGGCCACTTGATC]
Rev [CCATCTCTTGGGCCTCTTTCT]	Delivery of tRNA for protein synthesis	85	59	2.00	0.998	MH687364	
Lst-GAPDH	For [CAACAACCGACAAAGCAA]
Rev [CATAACAAACATAGGGGCA]	Carbohydrate metabolism	93	55	1.82	0.988	MH687363	
Lst-Rapac1	For [GGCTCTTTCTTTCCCTTTGT]
Rev [TTCCTGCTCTTCTTGCGT]	Cellular trafficking	124	58	1.82	0.989	MH687365	
Lst-TUBB	For [GGCTAGGGGATGAAGATGA]
Rev [AGGATGAGGGTGAATTTGA]	Microtubule element—cell structure	130	56	1.80	0.993	KX387887	
Lst-UBI	For [GTATTGTGGTGCTGGTGTTTT]
Rev [GCTTCCTCCTCTGGTTTGT]	Regulate protein function	105	59	1.94	0.993	MH687367	
Lst-VGKC	For [TGGCTTCCTGCTTCTCTGT]
Rev [GCTTCTGTCGTTGTTTTTGCT]	Maintenance of cell membrane potential	99	60	1.82	0.997	MH687366	

After the optimal primer sets were selected based on reaction specificity, primer efficiencies were calculated for each primer set. Primer efficiency curves were generated from RT-qPCR reactions on serial dilutions of RNA. The parent reaction contained 100 ng total RNA and four 1:5 dilutions were performed to generate five Cq values.

Quantitative PCR

The minimum information for publication of quantitative real-time (MIQE) PCR experiments guidelines were followed throughout the collection of qPCR data (Bustin et al., 2009). Amplification of all genes was detected with SyBR Green dye which generates fluorescence based on the synthesis of double-stranded DNA. The reactions contained two µL of cDNA with 10 µL of Bio-Rad SsoAdvanced Universal SyBR Mix, 600 nM forward and reverse primer concentration, and topped to 20 µL with DEPC H2O. Each replicate of L. stagnalis tissue was subjected to qPCR reactions in triplicate. The qPCR reactions took place in a Bio-Rad CFX Connect thermocycler running a custom program. The custom qPCR program consisted of 95 °C for 30 s; 40 cycles of 95 °C for 15 s, 55 °C for 30 s. The plate was read by the machine to measure fluorescence at the end of each cycle.

Data analysis

The expression stability of each gene was assessed with four computational algorithms: geNorm (Vandesompele et al., 2002), NormFinder (Andersen, Jensen & Ørntoft, 2004), BestKeeper (Pfaffl et al., 2004), and RefFinder (Xie et al., 2012). GeNorm ranks candidates by their expression stability (M) values that are assigned following pair-wise variation measurements among genes. Genes with M < 1.5 are considered to be stable. The NormFinder algorithm compares intra-group variation (i.e., mRNA levels of one gene within the tissue of interest) to inter-group variation (i.e., mRNA levels of other genes in the same tissues) and assigns genes a stability value based on variation among Cq values. BestKeeper builds a stability index based on repeated pair-wise correlation analyses between every reference gene and judges reference gene stability based on the standard deviation (SD) from the index, genes that have a smaller SD will be ranked more highly by BestKeeper. Finally, RefFinder was used to combine the ranked results from each algorithm and assign each gene an overall rank.

Data analyses were performed to compare the stability of among candidate reference genes in each tissue of interest. Following the tissue-specific analyses, we also combined all data for analysis together to simulate the conditions of a whole-body analysis. This approach allowed for the identification of which genes are most stable in each tissue of interest and provided evidence for which genes would be the strongest reference genes in an RT-qPCR experiment that used RNA from a whole L. stagnalis body. Given that a minimum of two reference genes are typically recommended for proper RT-qPCR data normalization, the recommended reference genes are generally expressed in terms of pairs.

Results

Primer specificity and efficiency

Primers for all candidate reference genes were evaluated to ensure that they could produce consistent results and not amplify off-target products or generate primer dimers. Following amplification, each primer pair produced amplicons that yielded single bands at the correct size after electrophoresis in 2% agarose gels (Fig. 1). Additionally, no amplification was observed in controls that lacked reverse transcriptase in the RT-PCR or lacked cDNA template or primers in qPCR. Thus, primer pairs specifically amplified a single cDNA target. Based on the standard curves, primer set efficiencies ranged from 90% (GAPDH and Rapac1) to 106% (ACTB) with correlation coefficients (R2) of >0.980 (Table 1).

Figure 1 Representative image of PCR products for each reference gene.

A five µL sample of each PCR product was run on a 2% agarose gel that also contained a no primer control (NPC).

Tissue-specific expression profiles and stability of reference genes

The candidate reference genes showed a variety of expression levels across individual tissues (Fig. 2). Lst-UBI produced the smallest mean Cq values and displayed the smallest Cq ranges for most tissues (Table 2). Conversely, Lst-VGKC mRNA was the least abundant in every tissue and displayed large Cq ranges.

Figure 2 Expression levels for each gene in each tissue based on Cqs.

Data was collected from three replicates of L. stagnalis tissue and each reaction was performed in triplicate. The tissues examined were: (A) CNS, (B) tentacles, (C) lips, (D) penis, (E) foot, and (F) mantle. Box plot: upper and lower box limits indicate 25th and 75th percentiles, line of division in box indicates the median, and whiskers indicate the minimum/maximum values.

Table 2 Mean Cq values and standard errors of the mean (SEM) of reference genes for tissues in Lymnaea stagnalis.

Means were calculated from three replicates of L. stagnalis tissue, each reaction was performed in triplicate.

Tissue	Lst-ACTB	Lst-EF1α	Lst-GAPDH	Lst-Rapac1	Lst-TUBB	Lst-UBI	Lst-VGKC	
Cq mean	SEM	Cq mean	SEM	Cq mean	SEM	Cq mean	SEM	Cq mean	SEM	Cq mean	SEM	Cq mean	SEM	
CNS	19.27	0.67	19.49	0.53	21.10	0.56	26.81	0.81	18.30	0.76	16.72	0.20	29.27	0.82	
Tentacles	17.42	0.20	18.50	0.22	20.14	0.21	27.36	0.43	17.99	0.36	16.41	0.19	28.86	0.23	
Lips	18.19	0.27	19.14	0.23	21.01	0.21	26.97	0.19	18.39	0.29	17.11	0.27	29.26	0.28	
Penis	18.30	0.38	17.87	0.25	19.71	0.25	26.44	0.35	22.05	1.98	16.55	0.16	30.60	0.23	
Foot	18.08	0.41	18.91	0.48	20.22	0.46	25.72	0.46	18.53	0.25	16.69	0.24	30.48	0.17	
Mantle	17.67	0.26	18.91	0.24	20.41	0.34	27.57	0.39	22.16	0.42	16.24	0.16	30.78	0.31	
Overall	18.15	0.17	18.80	0.15	20.43	0.16	26.81	0.20	19.57	0.43	16.62	0.09	29.87	0.19	

The algorithms used to measure the stability of each candidate reference gene were highly congruent in their tissue-specific rankings. Therefore, only the results from the RefFinder analysis (which combines the rankings from the other three algorithms) are presented in Fig. 3 (results for other algorithms reported in Figs. S1–S3). Lst-GAPDH and Lst-EF1α were the most stable candidate reference genes in the CNS and penis. Lst-UBI and Lst-GAPDH were the most stable in the tentacles, Lst-GAPDH and Lst-Rapac1 were the most stable in the lips, Lst-UBI and Lst-ACTB were the most stable in the foot, and Lst-UBI and Lst-VGKC were the most stable in the mantle. Overall, identification of the most reliable candidate reference gene was tissue dependent. However, according to geNorm, all genes except for Lst-ACTB and Lst-TUBB were acceptable as a reference genes for all tissues as they met the minimum stability threshold of M < 1.50 for heterogeneous tissue samples.

Figure 3 RefFinder rankings for all candidate reference genes by tissue.

RefFinder rankings for the (A) CNS, (B) tentacles, (C) lips, (D) penis, (E) foot, and (F) mantle. RefFinder calculates rankings as the geometric mean of the rankings assigned by geNorm, NormFinder, and BestKeeper. Genes are ranked in order from the least stable to the most stable in each panel.

Expression levels and stability of reference genes in a combined tissue analysis

To assess which candidate reference genes could be the most stable in whole-body preparations of L. stagnalis, the data from all tissues was combined and analyzed together with geNorm, NormFinder, BestKeeper, and RefFinder. The candidate reference genes spanned a wide range of expression levels when data from all tissue types was combined. Based on the Cq values, Lst-UBI was the most highly expressed gene (mean Cq 16.62 ± 0.09) and had the smallest overall Cq range whereas Lst-VGKC was the least expressed gene and had the highest biological variability as measured by the standard error of the mean (mean Cq 29.87 ± 0.19; Table 2). All four algorithms used to assess expression stability were highly congruent in their ranking of candidate reference genes in the combined analysis (Fig. 4). GeNorm, NormFinder, and RefFinder rated Lst-GAPDH and Lst-EF1α as the top two most stable reference genes. BestKeeper ranked Lst-UBI as the most stable with Lst-EF1α and Lst-GAPDH as the second and third most stable, respectively.

Figure 4 Combined analysis rankings assigned to each candidate reference gene.

Data from all tissues was combined and assigned ranks by (A) geNorm, (B) NormFinder, (C) BestKeeper, and (D) RefFinder. *Indicates that both genes were equally recommended.

Discussion

We employed several established algorithms to analyze mRNA abundance data and identify stably expressed genes that are suitable as reference genes for RT-qPCR in L. stagnalis. We assessed seven candidate reference genes (Lst-ACTB, Lst-EF1α, Lst-GAPDH, Lst-Rapac1, Lst-TUBB, Lst-UBI, and Lst-VGKC) in six tissues of interest (CNS, tentacles, lips, penis, foot, and mantle) and provided a separate analysis of all tissues combined. There were variable patterns of expression stability among the genes in the different tissues, but the tissue-specific rankings produced by the different algorithms were highly congruent. Lst-GAPDH and Lst-EF1α were the most suitable pair of reference genes in the CNS and penis. However, Lst-UBI also was one of the two most stable genes the tentacles, foot, and mantle when paired with Lst-GAPDH, Lst-ACTB, and Lst-VGKC, respectively. Finally, Lst-GAPDH and Lst-Rapac1 were the most stable in the lips. In the combined analysis, Lst-GAPDH, Lst-EF1α, and Lst-UBI were the most stable genes according to all algorithms. Thus, for future experiments using either whole animals or tissues or tissue combinations not included in our analysis, we recommend these three genes as the first candidates for validation as reference genes.

Previous measurements of RT-qPCR reference gene stability has revealed some consistent patterns across the molluscs. EF1α appears to be the most effective reference gene in several mollusc species in addition to the CNS and penis of L. stagnalis (Table 3). In one study of L. stagnalis, EF1α was identified to remain stable in the CNS under heat stress (Foster, Lukowiak & Henry, 2015). Given that EF1α has also been identified to be highly stable in several other molluscs, it appears to be a strong candidate reference gene across the phylum (Morga et al., 2010; Wan et al., 2011; Cubero-Leon et al., 2012; Mauriz et al., 2012; Moreira et al., 2014; García-Fernández et al., 2016; Huan, Wang & Liu, 2016). An analysis of reference genes in the freshwater snail Bellamya aeruginosa showed that EF1α and GAPDH were stable in the tentacles and penis but more variable in the foot (Liu et al., 2015), similar to our findings (albeit with some discrepancies of rankings in the tentacles and foot). GAPDH has also been shown to be a stable reference gene in bivalves (Morga et al., 2010; Martínez-Escauriaza et al., 2018) but was reported as unsuitable in abalone and octopus (Wan et al., 2011; García-Fernández et al., 2016). Thus, GAPDH is not consistent across the molluscs. Additionally, GAPDH is highly stable in the terrestrial gastropod C. nemoralis within a given season, but expression levels in this species are subject to great variation between seasons (Affenzeller, Cerveau & Jackson, 2018). In our data, Lst-UBI had a very consistent expression profile between tissues compared to other reference genes and was found to be highly stable in the foot, mantle, and tentacles as well as in the combined analysis. UBI has also shown promise in other molluscs (Sirakov et al., 2009; García-Fernández et al., 2016; Affenzeller, Cerveau & Jackson, 2018). Alpha-tubulin (TUBA) has generally produced positive results as a reference gene in several molluscan species (Sirakov et al., 2009; Cubero-Leon et al., 2012; Moreira et al., 2014). However, TUBA was unstable in C. nemoralis (Affenzeller, Cerveau & Jackson, 2018) and we also found Lst-TUBB to be highly unstable. ACTB has generally produced negative results as a candidate reference gene in molluscs (Cubero-Leon et al., 2012; Moreira et al., 2014; Liu et al., 2015; García-Fernández et al., 2016; Huan, Wang & Liu, 2016) and the results here from L. stagnalis are largely congruent with these previous findings, although Lst-ACTB was stable specifically in the foot. Finally, Rapac1 and VGKC were identified as suitable reference genes in C. nemoralis (Affenzeller, Cerveau & Jackson, 2018). These genes did show relative stability in our analyses of the mantle and lips, but were relatively unstable in the other tissues examined.

Table 3 Summary of the stability rankings of reference genes from studies conducted in molluscs.

Organism	Most stable gene(s)	Least stable gene(s)	Other genes tested	Reference	
Gastropods					
Haliotis discus	EF1α/RPL5	18S rRNA	ACTB, BGLU, CY, GAPDH, H2A, HPRT, SDHA, TUBB, UBC, CYP4	Wan et al. (2011)	
Bellamya aeruginosa	RPL7	ACTB	18S rRNA, EF1α, GAPDH, TUBB, H2A, DRP2	Liu et al. (2016)	
Cepaea nemoralis	EF1α/ACTB	GAPDH	DNARP, FIB3, GTP8, Rapac1, RNAP, TUBA, UBI, VGKC	Affenzeller, Cerveau & Jackson (2018)	
Lymnaea stagnalis	EF1α/GAPDH	ACTB/TUBB	VGKC, UBI, Rapac1	Present study	
Cephalopods					
Octopus vulgaris	TUBA/UBI	18S rRNA	16S rRNA, ACTB, EF1α, TUBA	Sirakov et al. (2009)	
Octopus vulgaris	UBI	ACTB	18S rRNA, EF1α, GAPDH, TUBA	García-Fernández et al. (2016)	
Bivalves					
Ostrea edulis	EF1α/GAPDH	ACTB	UBI, RPL5	Morga et al. (2010)	
Mytilus edulis	EF1α/18S rRNA	ACTB	28S rRNA, TUBA, HEL	Cubero-Leon et al. (2012)	
Mytilus galloprovincialis	EF1α	ACTB	18S rRNA, TUBA	Moreira et al. (2014)	
Ruditapes philippinarum	TUBA	ACTB	18S rRNA, EF1α	Moreira et al. (2014)	
Crassostrea gigas	EF1α	RPS18	ACTB, ARF1, GAPDH, HNRPQ, UBC	Huan, Wang & Liu (2016)	
Mytilus galloprovincialis	GAPDH/RPS4	NAD4/18S rRNA	ACTB, COX1, GAPDH, RPS27, TIF5A	Martínez-Escauriaza et al. (2018)	
Note:

ARF1, adp-ribosylation factor 1; BGLU, beta-glucuronidase; COX1, cytochrome c oxidase subunit 1; CY, cyclophilin; CYP4, cytochromep450 family 4; DNARP, DNA repair protein; DRP2, DNA-directed RNA polymerase II; FIB3, fibronectin type III domain containing protein; GTP8, GTP-binding protein; H2A, histone H2A; HEL, RNA helicase; HNRPQ, heterogeneous nuclear ribonucleoprotein q; HPRT, hypoxanthine phosphoribosyltransferase 1; NAD4, NADH dehydrogenase subunit 4; RNAP, RNA-directed DNA polymerase; RPL5, ribosomal protein L5; RPL5, ribosomal protein L7; RPS4, 40S ribosomal protein S4; RPS27, 40S ribosomal protein S27; SDHA, succinate dehydrogenase; UBC, ubiquitin-conjugating enzyme.

Despite the heterogeneity in top-ranked reference genes among the tissues, there was some consistency in expression stability across all tissues. Though Lst-EF1α, Lst-GAPDH, and Lst-UBI were not the top ranked in every tissue as judged by RefFinder, they were relatively stable in all tissues tested which is reflected in the stability values provided by geNorm. Thus, we recommend these genes for use in multi-tissue comparisons in L. stagnalis. In particular, Lst-EF1α and Lst-GAPDH are a favorable pair as they have the most similar Cq ranges. Importantly, despite the demonstrated stability of these genes under baseline conditions reported here, it would be crucial to directly verify their stability under any experimental conditions (Kozera & Rapacz, 2013).

Conclusion

The validation of stable reference genes is necessary for the acquisition of reliable gene expression data. Therefore, it is important to perform a species-specific verification of reference gene stability before undertaking RT-qPCR experiments. The variable expression of the seven genes among the six tissue types we investigated in L. stagnalis demonstrates that it is critical to select reference genes based on the tissues of interest. The results presented here should guide the selection of reference genes for tissue-specific RT-qPCR and thereby assist with future studies of gene expression in the snails.

Supplemental Information

Supplemental Information 1 GeNorm stability rankings for all candidate reference genes in each tissue.

Genes are ranked in order from the least stable to the top two most stable.

Click here for additional data file.

Supplemental Information 2 NormFinder stability rankings for all candidate reference genes in each tissue.

Genes are ranked in order from the least stable to the most stable.

Click here for additional data file.

Supplemental Information 3 BestKeeper stability rankings for all candidate reference in each tissue.

Genes are ranked in order from the least stable to the most stable.

Click here for additional data file.

Supplemental Information 4 Raw Cq values for all qPCR reactions.

Reactions were performed in triplicate for each gene and each tissue. For this dataset, 500 ng of RNA was used in the reverse transcription reaction prior to qPCR.

Click here for additional data file.

Supplemental Information 5 All Cq values for qPCR reactions.

This is an additional dataset not referred to in the current version of the article. All raw Cq values for every qPCR reaction. Reactions were performed in triplicate for each gene and each tissue. For this dataset, 20 ng of RNA was used in the reverse transcription reaction prior to qPCR.

Click here for additional data file.

We thank Dr. Scott Cummins for advice, the animal care staff at St. Francis Xavier University for maintaining the Lymnaea stagnalis colony, and both Hisayo Sadamoto and anonymous reviewer for constructive comments.

Additional Information and Declarations

Competing Interests

Author Contributions

DNA Deposition

Data Availability

The authors declare that they have no competing interests.

Alexander P. Young conceived and designed the experiments, performed the experiments, analyzed the data, prepared figures and/or tables, authored or reviewed drafts of the paper, approved the final draft.

Carmen F. Landry performed the experiments, analyzed the data, authored or reviewed drafts of the paper, approved the final draft.

Daniel J. Jackson contributed reagents/materials/analysis tools, authored or reviewed drafts of the paper, approved the final draft, supplied transcriptomes used to identify L. stagnalis gene sequences.

Russell C. Wyeth conceived and designed the experiments, contributed reagents/materials/analysis tools, authored or reviewed drafts of the paper, approved the final draft.

The following information was supplied regarding the deposition of DNA sequences:

The novel reference gene sequences (GAPDH, EF1a, Rapac1, UBI, VGKC) are available at GenBank: MH687363–MH687367.

The following information was supplied regarding data availability:

The raw data set is available at Figshare: Young, Alex (2019): Primary dataset—RefGene ALL Raw Cqs (500 ng).xlsx. figshare. Dataset. DOI 10.6084/m9.figshare.8329937.v1

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
