# Peer review of "Tissue-specific evaluation of suitable reference genes for RT-qPCR in the pond snail, Lymnaea stagnalis"

_PeerJ, doi:10.7717/peerj.7888_

## Round 0.1 · original submission · Major Revisions

· Academic Editor

Major Revisions

As you can see from the reviews, both reviewers emphasize the value your work has for the community. Stable reference genes are needed for the Lymnaea model organism and your work provides the necessary experiments for the community to build upon.

At the same time, both reviewers focus on the qPCR method. Importantly, Reviewer #2 points out that the sample size may be too low (due to the high variability), that cross-tissue pooling is not admissible and that PCR conditions need to be optimized.

While I am commonly very hesitant to suggest additional experiments, in this case these suggestions seem to be all very reasonable and, given the importance of this work for future research, very well worth the extra effort.

·

Basic reporting

no comment

Experimental design

no comment

Validity of the findings

no comment

Additional comments

The authors explore the stable reference genes for reverse transcription quantitative PCR (RT-qPCR) in the pond snail Lymnaea stagnalis. The seven reference genes were investigated across six different tissues with RT-qPCR. The stability of each of the genes was measured by some of the standard stability assessment algorithms. As a result, the authors described that GAPDH and EF1α are highly stable in the tissues as well as in pooled analyses. They also discussed those genes stability in other molluscs in the similar studies.
The main research findings of this paper will be important for the future researches in Lymnaea stagnalis and other molluscs. I would recommend it for acceptance after the minor points listed below.


Specific comments:
1. Line 56, 60, and 67
The authors referred RT-qPCR and in situ hybridization as the methods for investigating the gene expression. I am not sure that the part of in situ hybridization is necessary. The method RT-qPCR is to quantify the gene expression, and in situ hybridization is the method to identify the cells that express a particular gene. Northern blots, “Virtual Northern” blots with amplified cDNA and RNAseq are the proper methods for the quantification and comparison of gene expression.

2. Line 59
The product in RT-qPCR in Lymnaea stagnalis (Sadamoto et al., 2010), which should also be cited in this connection here.

3. Line 110
“10 - 50 mg” should not to be the weight of CNSs from two animals.

4. Line 119, 166 and 173
The quantity of total RNA in the cDNA sample should be described.

5. Line 131, 136 and 137
This section describes about reverse transcription, and not PCR.

6.Line 228
“lowliest” is misspelled.

7.Table 2 and 3
The titles of tables are both described as “table 1”.

8. Table 3, Discussion
The authors refer to the previous result that GAPDH in Cepaea nemoralis is stable, but variable between seasons. I cannot understand what the authors mean by these sentences. If the authors think that GAPDH result in different species (Table 3) come from the seasonal effects, they need to develop this idea further and explain it in Discussion.

Reviewer 2 ·

Basic reporting

No comment.

Experimental design

No comment.

Validity of the findings

No comment.

Additional comments

Evaluation of candidate reference genes for qRT-PCR experiments in the invertebrate experimental model Lymnaea stagnalis is lacking. It is of high importance therefore for quantification and comparisons in gene expression studies using qRT-PCR to validate the stability of candidate reference genes in different tissues and treatment groups in this species. In this manuscript, the authors evaluated a panel of seven reference genes in different tissues (central nervous system, tentacles, lips, penis, foot and mantle) in Lymnaea including 1-alpha (EF1), glyceraldehyde-3-phosphate dehydrogenase (GAPDH), beta-actin, beta-tubulin, ubiquitin, prenylated rab acceptor protein 1, and voltage gated potassium channel (VGKC) by qRT-PCR. To assess the stability of these genes in different tissues, the authors used the following standard stability assessment algorithms: geNorm, NormFinder, BestKeeper and RefFinder. GAPDH and EF1 were found to be highly stable in all of the tissues examined, however VGKC was found to be the least stable gene. Based on these results, the authors suggest that GAPDH and EF1 should be both used as reference genes for qRT-PCR studies in Lymnaea.
This is an important study because Lymnaea is a useful experimental invertebrate model used by several research groups working on learning and memory and it is necessary for qRT-PCR studies to use stable reference genes among tissues and treatments. The manuscript is very well and clearly written. However, there are some major issues that need to be addressed.
Major Revisions:
 The authors need to increase the N numbers as there was a huge variability in the Cq values for all the candidate reference genes examined including the GAPDH and EF1. The authors state that GAPDH and EF1a are highly stable and they should be used as reference genes for Lymnaea, however there was a high variability in their Cq values. The highest Cq value for GAPDH was 33 and the lowest was 20. The highest Cq value for EF1a was 30 and the lowest value was 24.
 For each candidate reference gene, the authors should optimise the different qRT-PCR conditions for each tissue individually as some Cq values were high for some tissues. For instance, the Mean of Cq value for Lst-EF1a was 27.43 in the Foot and 27.20 in the Mantle. The mean of Cq value for Lst-GAPDH was 26.62 in the foot and 25.75 in the Mantle. Cq values of higher than 25 are too high for a reference gene.
 Results section, Figure 1: For each candidate reference gene, graphs should be plotted for each tissue individually. As there is variability of gene expression of the candidate reference genes among tissues, data plots and analysis should not be pooled across all tissues. Furthermore, Individual data points should be plotted for each gene.
 Results section, Figure 2: Comprehensive rankings assigned by geNorm, NormFinder, BestKeeper and RefFinder should be performed for each individual tissue and not from pooled Cq values from all tissues.
 Conclusions on the rankings should be only drawn from Cq values from individual tissues and not from pooled Cq values from all the tissues as there is a huge variability of gene expression of the candidate reference genes among the different tissues.
Minor Revisions:
 Methods section, “Total RNA Extraction”, Line 105, “…animals were anesthetized in in 0.125%” should be changed to “…animals were anesthetized in 0.125%”.
 Methods section,” Reverse Transcription PCR section”, Line 135: “PCR reactions, 20 ng of RNA was added…” The authors must have made a typing error and they should change the 20 ng to 200 ng.
 Line 142. Nucleotides, not base pairs.
 Line 143. 57-62oC stated here for the melting temperature range but in Table 1 the temperature range is 55-60oC. Which one is correct?

---

## Round 0.2 · Minor Revisions

· Academic Editor

Minor Revisions

As you can see, there is only one minor detail to be added to the methods section to satisfy the last reviewer. This should not be difficult, I presume.

·

Basic reporting

no comment

Experimental design

Line 126
The authors used Lst-tyrosine hydroxylase to check the genomic DNA contamination in the RNA samples, however the reason was not described. The authors need to note the primer sequences and the reason to use it.

Validity of the findings

no comment

Additional comments

The authors explore the stable reference genes for reverse transcription quantitative PCR (RT-qPCR) in the pond snail Lymnaea stagnalis. The seven reference genes were investigated across six different tissues with RT-qPCR. The stability of each of the genes was measured by some of the standard stability assessment algorithms. As a result, the authors described that GAPDH and EF1α are highly stable in the tissues as well as in pooled analyses. They also discussed those genes stability in other molluscs in the similar studies.
The main research findings of this paper will be important for the future researches in Lymnaea stagnalis and other molluscs. The manuscript has been revised well. However I think that there is an improvement that should be made before publication.

Reviewer 2 ·

Basic reporting

No comment.

Experimental design

No comment.

Validity of the findings

No comment.

Additional comments

The authors have addressed all the concerns and have revised accordingly. Furthermore, the additional experiments performed by the authors improved the qRT-PCR results and the results are more robust as the Cq values for the candidate reference genes are lower and there is less variability. The manuscript is really well and clearly written. Validation of the stability of candidate reference genes for qRT-PCR experiments for specific tissues in the invertebrate experimental model Lymnaea stagnalis is important and the main research findings of this paper can be used for future experiments using Lymnaea stagnalis.

---

## Round 0.3 · Minor Revisions

· Academic Editor

Minor Revisions

There is still one small detail the final reviewer would like to have addressed. Please let me know if this issue poses a more severe problem that what it looked like at first.

·

Basic reporting

no comment

Experimental design

no comment

Validity of the findings

no comment

Additional comments

Lst-tyrosine hydroxylase is not the general reference gene. The authors added the explanation that "it is a gene of interest that is frequently amplified by us as a part of a larger research project on neural development". However, the Lst-tyrosine hydroxylase is highly expressed only in the CNS in many species, and it does not seem reasonable to use it to check the RNA quality in other tissues. It would be better to add reference papers of the authors, or, the authors should add the explanation that this gene is expressed in many tissues in Lymnaea stagnalis. The expression pattern of this gene in several tissues has not yet been reported in molluscan species, so it needs some explanation. Otherwise, the amplifications in many tissues seem possibly to be due to the sample contamination.

---

## Round 0.4 · accepted · Accept

· Academic Editor

Accept

Congratulations, the final reviewer is happy with your explanations now. Please see their final comments.

·

Basic reporting

no comment

Experimental design

no comment

Validity of the findings

no comment

Additional comments

As the authors wrote, I misunderstood the word "positive control", and now I understand the experiment to check the genomic DNA contamination. However, I still do not understand the primer design. The authors had better to add the explanation about the primers that they can be used for genomic DNA amplification, or that the authors usually use those primers for that purpose. The genomic DNA sample can be used to check the reliability of those primer design. It is not necessary to say the primers can be used for RT-PCR.
The genomic DNA sequence of Lymnaea is not available now. But, as the exon-intron structure are generally well conserved, I checked molluscan species genomic data. The primers of LymEF1a seem to be designed in the same exon, which means the primers can amplify the genomic DNA. Even though this is not the direct evidence, the primer design seems to be reasonable.